# Taste Dysfunction in Head and Neck Cancer: Pathophysiology and Clinical Management—A Comprehensive Review

**DOI:** 10.3390/biomedicines13081853

**Published:** 2025-07-30

**Authors:** Luigi Sardellitti, Enrica Filigheddu, Giorgio Mastandrea, Armando Di Palma, Egle Patrizia Milia

**Affiliations:** 1Department of Medicine, Surgery and Pharmacy, University of Sassari, 07100 Sassari, Italy; g.mastandrea1@studenti.uniss.it (G.M.); emilia@uniss.it (E.P.M.); 2Dental Unit, Head and Neck Department, Azienda Ospedaliero Universitaria, 07100 Sassari, Italy; enrica.filigheddu@aouss.it; 3Department of Emergency, Admission, Anesthesia and Critical Care, Azienda Ospedaliero-Universitaria Policlinico Umberto I, 00161 Rome, Italy; a.dipalma@policlinicoumberto1.it

**Keywords:** head and neck cancer, taste dysfunction, dysgeusia, chemotherapy, radiotherapy, supportive care

## Abstract

**Background/Objectives**: Taste dysfunction is a highly prevalent yet underrecognized complication among patients with head and neck cancer (HNC), significantly impairing nutritional intake, treatment adherence, and quality of life (QoL). This comprehensive review synthesizes current knowledge on the pathophysiological mechanisms and clinical management of taste dysfunction associated with HNC and its treatments, particularly chemotherapy and radiotherapy. **Methods**: A structured literature search was performed across PubMed, Scopus, and Cochrane Library for articles published between January 2015 and February 2025. Studies were included if they investigated taste dysfunction related to HNC, focusing on pathophysiological mechanisms and therapeutic interventions. A total of 47 original studies were analyzed through a narrative synthesis due to heterogeneity in study designs and outcomes. **Results**: Taste dysfunction in HNC patients arises from tumor-related inflammation, cytotoxic injury from chemotherapy, and radiation-induced epithelial and neural damage. Chemotherapy and radiotherapy often exert synergistic negative effects on gustatory function. Management strategies identified include dietary counselling, nutritional supplementation (zinc, lactoferrin, monosodium glutamate, miraculin), pharmacological agents targeting salivary function, and non-pharmacological interventions such as acupuncture, photobiomodulation, and reconstructive surgery. However, the evidence is limited by small sample sizes, methodological variability, and the frequent exclusion of HNC patients from broader dysgeusia trials. Reported prevalence of taste dysfunction ranged from 39% to 97.4%**,** with higher rates observed among patients treated with radiotherapy and chemoradiotherapy. **Conclusions**: Taste dysfunction remains a critical yet unmet clinical challenge in HNC patients. High-quality, targeted research is urgently needed to develop standardized assessments and evidence-based management strategies to improve patient outcomes.

## 1. Introduction

Taste is the perception resulting from the stimulation of taste receptors in the oral cavity by chemical molecules. This sensory process, essential for detecting safe or harmful foods, works in synergy with smell, sight, and even hearing, contributing to appetite regulation and the initiation of digestive functions [1,2,3]. Taste receptors are specialized neuroepithelial cells organized in clusters known as taste buds, which are found not only on the tongue—within fungiform, circumvallate, and foliate papillae—but also in the epiglottis, pharynx, and larynx [2,3]. The lifespan of these cells ranges from 8 to 12 days but can extend up to 24 days in certain conditions [4]. Taste perception involves five basic qualities—sweet, salty, sour, bitter, and umami—mediated by G protein-coupled receptors and ion channels. While sweet, umami, and bitter stimuli are recognized by Type II taste receptor cells (T1R family), sour and salty tastes are mediated by Type III and Type I cells, respectively. Notably, Type I cells are the most abundant and likely contribute to maintaining structural integrity and regulating synaptic transmission. Gustatory signals are ultimately transmitted to the central nervous system via cranial nerves, including the facial, glossopharyngeal, vagus, and trigeminal nerves [2,3,5].

Saliva plays a crucial role in oral health and, more specifically, in taste perception. It is composed of 99% water and less than 1% solid components, including electrolytes and proteins. Secreted by major (parotid, submandibular, sublingual) and minor salivary glands, its production is tightly regulated by the autonomic nervous system and is closely linked to masticatory and gustatory reflexes [6,7,8].

Saliva enables the solubilization and transport of tastants, preserves ionic homeostasis around taste receptors, and protects the taste buds from chemical and mechanical injury. Furthermore, peptides such as leptin, ghrelin, insulin, neuropeptide Y (NPY), and peptide YY (PYY) may modulate gustatory signaling [3]. Variations in salivary flow rate significantly influence the accuracy of taste perception, particularly for salty and sour stimuli, while sweetness and bitterness are less affected by such changes. Therefore, both qualitative and quantitative salivary alterations—caused by systemic conditions, pharmacologic treatments, or local damage—can result in clinically relevant taste impairment.

This interplay becomes particularly critical in patients with head and neck cancer (HNC), a heterogeneous group of malignancies affecting the oral cavity, pharynx, larynx, nasal passages, and salivary glands. These tumors represent a substantial global burden, with incidence increasing in some regions due to shifts in tobacco and alcohol consumption and human papillomavirus (HPV) infection [6]. Among the multiple adverse effects experienced by HNC patients, taste dysfunction (reported in 17.6% to 93%) and xerostomia (40.4% to 93%) are particularly prevalent, both before and after treatment [7,8,9]. Taste disturbances—ranging from partial (dysgeusia) to complete loss (ageusia)—may compromise dietary intake, reduce adherence to therapy, and deteriorate quality of life. In more severe cases, they are associated with weight loss, emotional distress, and premature treatment discontinuation [8,9,10].

The causes of taste impairment in HNC are multifactorial. Cancer-related inflammation, direct tumor involvement, surgical resection, radiotherapy, and chemotherapy can all contribute to the dysfunction. These effects may damage taste receptors, alter cranial nerve transmission, and disrupt salivary quantity or composition. The physiological dependence between taste and saliva further amplifies the consequences of any single alteration [3]. Despite the frequency and clinical relevance of these complications, taste dysfunction remains underrecognized compared with other treatment-related toxicities such as mucositis or xerostomia. Its pathophysiology is still not fully understood, and major clinical guidelines from oncologic societies like NCCN, MASCC/ISOO, and ESMO do not provide specific recommendations for its prevention or treatment.

Given these clinical and scientific gaps, this review aims to provide an updated and comprehensive synthesis of the evidence on taste dysfunction in HNC patients, with particular attention to pathophysiological mechanisms and current management strategies.

## 2. Materials and Methods

### 2.1. Objective and Review Design

This comprehensive review explores the pathophysiology and clinical management of taste dysfunction in patients with HNC, particularly focusing on dysgeusia caused by the tumor itself, and the associated chemotherapy and/or radiotherapy. The objectives are to: (1) examine how HNC and its treatments affect taste perception; (2) summarize current interventions—including dietary, pharmacological, and non-pharmacological approaches—aimed at mitigating dysgeusia; and (3) highlight research gaps, especially regarding the difficulty of isolating taste-related outcomes in head-and-neck-specific populations. The review was conducted using a transparent and structured methodology, inspired by the PRISMA-ScR (Preferred Reporting Items for Systematic Reviews and Meta-Analyses extension for Scoping Reviews) guidelines, which are specifically designed to improve the transparency and reporting of exploratory reviews [11]. The research question was framed using the Population–Concept–Context (PCC) model [12], considering patients with head and neck or oropharyngeal cancer as the population, taste dysfunction (including dysgeusia, hypogeusia, and ageusia) as the concept, and pathophysiological mechanisms and clinical interventions as the context.

### 2.2. Literature Search Strategy

A structured literature search was performed across PubMed (MEDLINE), Scopus, and the Cochrane Library, covering publications from January 2015 to February 2025. The search was last updated in March 2025. The search strategy employed a combination of MeSH terms and free-text keywords, using Boolean operators to combine terms such as “taste dysfunction”, “dysgeusia”, “head and neck cancer”, “oropharyngeal cancer”, “radiotherapy”, “chemotherapy”, “treatment”, and “intervention” (Appendix A). No restrictions were applied regarding language or study design. Additional studies were identified through manual screening of reference lists.

### 2.3. Study Selection Process

All retrieved records were imported into Rayyan QCRI (Rayyan Systems Inc., Doha, Qatar), a web-based platform for systematic review management. After removing 735 duplicates, 1225 articles remained for title and abstract screening. Two reviewers independently assessed all records for eligibility, with discrepancies being resolved through discussion. Following this process, 68 full-text articles were assessed, and 47 original studies were finally included in the review. The study selection process is illustrated in the PRISMA 2020 flow diagram (Figure 1).

### 2.4. Eligibility Criteria

Studies were included if they enrolled human participants with head and neck or oropharyngeal cancer, reported taste dysfunction either caused by the tumor itself or induced by chemotherapy and/or radiotherapy, and provided clinical, mechanistic, or therapeutic data specifically related to taste alterations. Eligible study designs included randomized controlled trials, prospective and retrospective cohort studies, observational studies, and pilot clinical trials. Systematic and narrative reviews were consulted to contextualize findings but were not included in the primary synthesis. Studies were excluded if they were preclinical (animal or in vitro studies), focused exclusively on olfactory dysfunction, investigated non-cancer-related taste alterations, or were limited to abstracts, letters, or editorials without available full-text articles.

### 2.5. Data Extraction and Synthesis

Data were manually extracted using a standardized Excel spreadsheet, piloted on a sample of included studies. Extracted data encompassed authorship, year, country, study design, tumor location, treatment modalities, taste dysfunction characteristics, assessment methods, and any therapeutic interventions investigated. Studies were thematically categorized into four major domains: taste dysfunction caused by the tumor itself, chemotherapy-induced taste dysfunction, radiotherapy-induced taste dysfunction, and clinical management of taste dysfunction. The clinical management section was further subdivided into nutritional, pharmacological, and non-pharmacological interventions. Due to heterogeneity in study designs and outcome measures, a narrative synthesis approach was adopted to integrate and interpret the findings.

### 2.6. Quality Considerations

Although this was a comprehensive but non-systematic review, qualitative appraisal of study quality was undertaken to support a balanced interpretation of findings. A formal risk-of-bias assessment tool (e.g., ROB 2.0, Newcastle–Ottawa Scale) was not applied due to the heterogeneity of study designs, outcomes, and assessment tools, which precluded meaningful comparison. Nonetheless, included studies were critically appraised based on key methodological features such as (1) study design (prospective > retrospective > cross-sectional); (2) sample size and representativeness; (3) use of validated tools for taste assessment (e.g., taste strips, electrogustometry, CiTAS); (4) timing of outcome measurement; and (5) reporting of confounders (e.g., mucositis, xerostomia, smoking). Greater interpretative weight was given to randomized controlled trials and prospective observational studies that reported clear inclusion criteria and standardized outcome measures. Conversely, findings from small, single-arm, or poorly described studies were interpreted with appropriate caution and clearly flagged in the narrative synthesis.

## 3. Results

The literature highlights multiple, interacting factors contributing to taste dysfunction in head and neck cancer patients, both before and after oncological treatments. These alterations can significantly impair nutritional intake, quality of life, and treatment adherence. Findings have been grouped into three main categories: (1) dysfunction caused by the cancer itself; (2) dysfunction related to chemotherapy; and (3) dysfunction resulting from radiotherapy. Although often interrelated, taste and salivary alterations stem from distinct pathophysiological mechanisms and may require tailored clinical approaches.

### 3.1. Taste Dysfunction Caused by Cancer Itself

Taste dysfunction can emerge early in the course of oropharyngeal cancer, even before any oncological treatment, with symptoms ranging from metallic taste to complete loss. These alterations may compromise eating behavior, nutritional intake, and quality of life [10,13]. Cancer-related inflammation plays a pivotal role in dysgeusia pathogenesis. Activation of Toll-like receptors (TLRs) in taste receptor cells triggers the release of pro-inflammatory cytokines (e.g., IFN-γ, IL-1β, IL-6, TNF-α), which promote apoptosis, impair taste cell turnover, and interfere with both peripheral and central gustatory processing [2,14]. These pathways are also implicated in cancer cachexia, which further contributes to taste impairment via systemic inflammation and nutritional deficiencies (e.g., zinc, vitamin B12) [15].

Although mechanistic links are well supported, only a few clinical studies have explored taste dysfunction in treatment-naïve patients. Singh et al. [16] observed significant sweet and salty taste loss associated with perineural invasion and tongue depapillation in early-stage oral cancer. Lilja and coworkers [17] found baseline taste threshold asymmetries favoring the healthy side using electrogustometry. Further evidence comes from Uí Dhuibhir et al. [18], who reported that 74% of patients with solid tumors (including HNC) experienced taste or smell alterations before treatment, with taste more frequently being affected. Cunha et al. [19] noted that 39% of advanced HNC patients had hypogeusia, especially for bitter taste, with T4 tumors increasing the risk by 2.27 times. This association between tumor burden and taste alteration was further corroborated by Abbas et colleagues [20], who confirmed lower taste-related quality of life scores in advanced-stage OSCC patients (*p* = 0.045).

#### Surgical Resection as an Independent Contributor to Taste Dysfunction

In addition to the pathophysiological consequences of tumor burden, surgical resection itself constitutes a primary and often irreversible cause of taste dysfunction in patients with head and neck cancer.

Resection of key anatomical structures—particularly the anterior two-thirds of the tongue, soft palate, and floor of the mouth—can markedly impair gustatory function due to direct damage to taste receptor fields and the disruption of associated neural pathways. The chorda tympani and lingual nerves are especially vulnerable during glossectomies and surgeries involving the floor of the mouth, leading to compromised signal transmission from the fungiform papillae [21]. Clinical evidence indicates that resections involving more than 50% of the mobile tongue are significantly associated with reduced perception of sweet and salty stimuli. These impairments frequently persist despite reconstructive efforts [22]. Moreover, the extent and symmetry of the surgical defect appear to play a critical role, as asymmetric resections may interfere with the bilateral integration of gustatory input and further exacerbate sensory dysfunction [5]. These observations highlight surgical excision of gustatory structures as a direct and independent contributor to dysgeusia, beyond tumor-induced inflammation and other oncologic treatments.

Altogether, the evidence suggests that tumor extension, perineural invasion, local inflammation, and surgical resection—independent of therapeutic interventions—can significantly disrupt gustatory function through neural involvement and damage to receptor zones. Key study data are summarized in Table 1.

### 3.2. Taste Dysfunction Caused by Chemotherapy

Chemotherapy-induced taste alterations (CiTAs) affect up to 70% of cancer patients and often emerge within 3–5 days of treatment initiation. Symptoms typically resolve within three weeks, although in some cases they may persist for months. Dysgeusia frequently coexists with mucositis, fatigue, nausea, and appetite loss, collectively worsening nutrition and quality of life [46,47,48].

The pathophysiology of CiTAs involves multiple mechanisms, including direct cytotoxicity to taste receptor cells, cranial nerve inflammation (VII, IX, X), salivary dysfunction, and zinc depletion. Taste cells, with a turnover of ~10 days, are highly sensitive to chemotherapeutic agents, especially those inhibiting the Sonic Hedgehog (SHH) pathway critical for taste bud maintenance [46,49]. Chemotherapy-induced oxidative stress and zinc chelation further impair gustatory signaling by altering gustin activity [2,8].

Drug classes commonly implicated include platinum compounds (cisplatin, carboplatin), taxanes (docetaxel, paclitaxel), anthracyclines, and oral 5-FU analogs. Platinum agents are often linked to metallic taste and ageusia, while taxanes—particularly nab-paclitaxel—can cause persistent elevations in taste thresholds for multiple stimuli [50,51,52,53].

In HNC patients, isolating the effect of chemotherapy is challenging due to frequent use of combined therapies. This population is often underrepresented in broader trials due to the confounding impact of radiotherapy, mucositis, and surgical interventions. Furthermore, most available studies include mixed cancer cohorts and multimodal treatment protocols, making it difficult to isolate the specific contribution of chemotherapy to gustatory dysfunction in HNC. Nevertheless, emerging evidence indicates that CiTA in this group is both prevalent and clinically significant.

At the molecular level, Tsutsumi et al. [23] identified altered taste receptor expression (↓T1R3, ↑T2R5) in HNC patients undergoing chemoradiotherapy, correlating with dysgeusia and phantogeusia. Ihara et al. [24] reported reduced perception across all taste qualities by week six, with partial recovery at three months. In a retrospective cohort, Malta et al. [28] found dysgeusia more prevalent in patients treated with cisplatin and radiotherapy, highlighting the additive impact of multimodal treatments.

Although further research is needed to isolate the specific effects of chemotherapy on taste in HNC patients, current data confirm its clinical relevance. Summary data are included in Table 1.

### 3.3. Taste and Salivary Dysfunction Caused by Radiotherapy

Taste and salivary dysfunctions are common and impactful side effects of radiotherapy (RT) in head and neck cancer (HNC). Dysgeusia affects up to 96% of patients during or shortly after treatment [54,55,56], typically emerging within the first 2–3 weeks and peaking between weeks 5 and 7 [4,34]. While partial recovery may begin 4–5 weeks post-treatment, long-lasting alterations have been reported for months or even years [42,54].

RT-induced dysgeusia arises from direct epithelial damage, basal cell inhibition, and architectural disruption of taste buds [45,57,58]. Radiation to central nervous structures, such as the temporal lobe or nasopharynx, may also impair gustatory neural processing. Severity correlates with cumulative dose, particularly when ≥60 Gy is delivered to the anterior tongue or oral cavity [4,35].

Salivary dysfunction further exacerbates dysgeusia. Radiation-induced damage to salivary glands reduces flow rate and alters the ionic and protein composition of saliva, impairing tastant solubilization and delivery [58,59]. Gustin, a zinc-dependent salivary protein essential for taste bud maintenance, is also reduced post-irradiation [4]. Broader sensory changes, including reduced chemesthetic and textural perception, have also been reported [15].

Numerous studies have documented the timing and severity of RT-related dysgeusia. Most report early onset by week 2, with peak dysfunction between weeks 5 and 7 [27,30,33,40,43]. Although some symptoms such as phantogeusia may resolve within six months, basic taste often remains altered. Epstein et al. [25] noted persistent deficits two years after treatment, while Moroney et al. [30] observed grade 2 dysgeusia in over 25% of patients at 12 weeks post-IMRT. Various assessment tools—such as CiTAS, MDASI-HN, EORTC QLQ-H&N35, CTCAE, and psychophysical tests—have been used to monitor dysgeusia, consistently confirming its trajectory despite methodological heterogeneity. Persistent taste deficits are well documented, even with advanced techniques like IMRT.

Comparative studies suggest reduced short-term dysgeusia with IMPT over IMRT [30], while dose escalation via stereotactic radiosurgery (SRS) has been associated with higher toxicity, especially with single large fractions [33]. Galitis et al. [27] also reported worsened gustatory symptoms following post-operative RT or CCRT, persisting in some cases at three months.

Collectively, these findings confirm that RT-induced taste dysfunction in HNC is multifactorial and dose-dependent, involving epithelial injury, neural damage, and salivary alterations.

Among the studies reporting quantitative data, the prevalence of taste dysfunction ranged from 39% to 97.4% across treatment modalities. Higher rates were observed in patients treated with intensity-modulated radiotherapy (IMRT) combined with chemotherapy (97.4%) [30] or VMAT protocols (97%) [43]. Patients undergoing concurrent chemoradiotherapy (CRT) also showed a high prevalence of dysgeusia (50–88%). In contrast, studies involving non-treated patients reported lower rates (39–74%), while those treated with multimodal approaches including surgery showed intermediate values (53–93%). These findings highlight the significant burden of dysgeusia across treatment types and support the need for early preventive strategies.

A summary of the relevant studies is presented in Table 1.

### 3.4. Clinical Management of Taste Dysfunction in Oropharyngeal Cancer Patients

The clinical management of taste dysfunction in patients with HNC requires a comprehensive, patient-centered approach that integrates dietary adjustments, pharmacological interventions, non-pharmacological strategies, and novel therapies. Over the past decade, multiple approaches have been investigated, including dietary and nutritional interventions, nutritional supplementation, pharmacological treatments, and non-pharmacological modalities. Given the close physiological interplay between taste perception and salivary function, many interventions are capable of addressing both dysfunctions simultaneously.

#### 3.4.1. Dietary and Nutritional Interventions

Nutritional strategies represent a cornerstone in the management of taste dysfunction associated with HNC. Dietary counselling typically focuses on reducing aversive taste experiences while preserving adequate nutritional intake. Patients are often encouraged to avoid silverware and limit the consumption of foods with metallic or bitter notes—such as red meat, coffee, and tea—in favor of mildly flavored, high-protein alternatives like chicken, fish, eggs, and dairy products. Additional techniques reported to improve food palatability include marinating meats, consuming cold or lukewarm dishes, flavor enhancement with herbs or sugar, increasing water intake, and adopting smaller, more frequent meals [2,47]. Lemon juice, chewing gum before meals, and oral rinses have been reported to stimulate appetite and improve taste perception. Dietary recommendations tailored to the specific taste alteration—such as cold foods for metallic taste or fractionated meals for bitter dysgeusia—may further enhance palatability and nutritional intake [51]. In parallel, maintaining optimal oral hygiene plays a key role in improving taste perception. Chlorhexidine mouthwashes, soft toothbrushes, and sodium bicarbonate rinses may help restore a healthier oral environment, thereby improving gustatory function [5,8,51].

##### Nutritional Supplements

Several nutritional supplements have been explored to mitigate taste dysfunction in HNC patients. Zinc—administered as zinc sulfate, polaprezinc, or elemental zinc—has received the most attention due to its key role in gustin synthesis and the regeneration of taste buds [51,57]. Prophylactic zinc sulfate (50 mg TID) was shown to significantly preserve sweet and salty taste during chemoradiotherapy in a randomized trial by Khan et al. [60]. However, its efficacy appears limited when dysgeusia is already established, likely due to irreversible damage to taste progenitor cells [57,61].

Lactoferrin, a salivary glycoprotein with antioxidant and anti-inflammatory activity, has shown promise in chemotherapy-induced taste and smell abnormalities. In a pilot study, daily lactoferrin (750 mg) significantly improved TSQ scores over 30 days, with sustained benefits one month after discontinuation [62]. However promising, these findings have yet to be validated in patients with head and neck cancer.

Similarly, monosodium glutamate (MSG) has been explored as a supportive intervention in HNC patients receiving chemoradiotherapy. Shono et al. [63] found that MSG supplementation during chemoradiotherapy preserved T1R3 expression and improved taste sensitivity and energy intake, suggesting both symptomatic and molecular benefits.

Miraculin, a glycoprotein extracted from Synsepalum dulcificum (commonly known as miracle fruit), has also gained attention for its unique taste-modifying properties. In acidic environments, miraculin binds to sweet taste receptors, eliciting a sweet sensation that temporarily enhances the palatability of sour or metallic foods [51,64]. This mechanism makes it particularly suitable for cancer patients experiencing dysgeusia, who often develop aversions to previously tolerated foods. A randomized trial by López-Plaza et al. [65] showed that miraculin supplementation reduced food aversion, improved appetite, and increased caloric intake in malnourished cancer patients with chemotherapy-induced dysgeusia.

The main characteristics and outcomes of these studies are presented in Table 2.

#### 3.4.2. Pharmacological Treatments

To date, no pharmacological agents have been specifically approved or universally recommended for the treatment of taste dysfunction in patients with head and neck cancer (HNC). The existing body of literature is largely composed of small-scale clinical trials, pilot studies, or observational reports, which have investigated a range of pharmacological strategies—including cytoprotective agents such as amifostine and salivary stimulants like pilocarpine. However, none of these interventions have demonstrated consistent, high-quality evidence supporting their efficacy in targeting dysgeusia as a primary outcome. First-line pharmacological treatments for radiation-induced xerostomia include pilocarpine and cevimeline, two muscarinic receptor agonists that enhance salivary flow in patients with preserved salivary gland function. Nevertheless, their effect on taste recovery remains variable and poorly characterized. In addition, adverse effects such as sweating, nausea, and gastrointestinal discomfort may significantly affect patient compliance and limit their long-term tolerability [79].

Despite incremental advances in supportive care, the current pharmacological landscape remains notably underdeveloped. Most available studies lack the methodological rigor of randomized controlled trials (RCTs) and are characterized by small sample sizes and short follow-up durations. Moreover, the heterogeneity in outcome measures—ranging from subjective questionnaires to non-standardized taste assessments—limits the ability to compare findings across studies. Issues related to patient adherence, particularly in the context of polypharmacy and treatment-related fatigue, further hinder the generalizability of results. These limitations underscore the need for robust, well-powered clinical trials specifically designed to evaluate the efficacy, safety, and tolerability of pharmacologic agents for dysgeusia in HNC populations.

#### 3.4.3. Non-Pharmacological Interventions

Non-pharmacological approaches represent valuable adjuncts in the management of taste dysfunction among patients with head and neck cancer, particularly when pharmacologic or dietary strategies prove insufficient or are poorly tolerated. These interventions are often preferred for their safety, minimal invasiveness, and potential to enhance quality of life. Among the most promising strategies are surgical reconstruction techniques and a growing array of complementary and integrative therapies (CIMs), which are increasingly recognized within supportive cancer care.

##### Surgical Reconstruction and Gustatory Preservation

Reconstructive approaches may influence long-term gustatory outcomes. Lu et al. [71] reported preserved taste function in all 21 patients treated with a modified anterior–posterior tongue rotation flap after T2N0 oral tongue carcinoma resection. These results suggest that flap design and anatomical symmetry may play a role in sensory preservation. In contrast, other studies—such as those by Li et al. [72], Yuan et al. [73], and Yue et al. [74]—did not report significant differences in taste outcomes across various flap techniques (RFFF, PMMF, ALTFF), indicating that surgical resection itself may be the primary determinant of long-term taste loss.

##### Complementary and Integrative Therapies

Various complementary approaches have been investigated for their potential to support taste and salivary function in HNC patients, including acupuncture, transcutaneous electrical nerve stimulation (TENS), photobiomodulation therapy (PBM), and anti-inflammatory natural products.

Acupuncture may improve dysgeusia via neuroprotective and anti-inflammatory pathways involving β-endorphins and cytokine modulation. Case reports and small prospective studies, such as those by Djaali et al. [75] and Ben-Arye et al. [66], described improved taste function and pain relief without adverse effects.

Neuromodulation techniques, such as TENS, have also shown potential. Dalbem Paim et al. [69] reported increased salivary flow and improved taste perception in 12 patients following bilateral TENS therapy over major salivary glands.

PBM, using a 635 nm diode laser, has demonstrated regenerative effects in taste receptor structures. El Mobadder et al. [76] reported full recovery of taste in a patient after 10 PBM sessions, likely through mitochondrial stimulation and epithelial repair.

Topical and biocompatible agents have also been evaluated. Heiser et al. [67] reported improved taste and smell function, alongside reduced xerostomia, in 98 patients treated with a combined oral and nasal liposomal spray. Fernandes et al. [78] evaluated Brazilian organic propolis (BOP) in a randomized trial, reporting lower inflammatory markers and milder symptoms, although without statistically significant differences in dysgeusia severity.

Intraoral shielding stents during radiotherapy, tested by Yangchen et al. [77], reduced mucositis and xerostomia but had no measurable effect on taste preservation.

While findings are encouraging, most derive from small-scale studies with limited generalizability. A detailed summary of these interventions is provided in Table 2.

## 4. Discussion

Taste dysfunction is a highly prevalent yet frequently underestimated complication among patients with head and neck cancer (HNC). This comprehensive review underscores its multifactorial etiology, encompassing both tumor-related mechanisms and treatment-induced insults—particularly those associated with chemotherapy and radiotherapy. Reported prevalence rates vary widely across studies, largely influenced by assessment timing, treatment modality, and the methods used for taste evaluation.

This review confirms a broad prevalence range of dysgeusia (39% to 97.4%) among HNC patients, with the highest values found in those undergoing radiotherapy-based regimens. Such heterogeneity reflects differences in treatment combinations, radiation techniques, and assessment methods, underscoring the need for standardized evaluation tools and tailored interventions.

Tumor burden alone can trigger early taste alterations, even before therapy initiation, primarily through inflammatory and neuroepithelial mechanisms. This dysfunction is mediated by a pro-inflammatory tumor microenvironment, where cytokines such as IL-1β, IL-6, IFN-γ, and TNF-α activate Toll-like receptors (TLRs), promoting taste bud apoptosis and impairing cellular turnover [2,14]. Additionally, cancer cachexia contributes to gustatory impairment via systemic inflammation, micronutrient deficiencies, and central sensory alterations [15]. Chemotherapy frequently induces acute taste changes, typically emerging within the first week of treatment. Mechanistically, this reflects direct cytotoxic effects on taste progenitor cells, oxidative stress, disruption of zinc homeostasis, and interference with the Sonic Hedgehog signaling pathway—particularly in patients treated with platinum-based agents and taxanes [23,25,29]. Radiotherapy, especially when targeting the oral cavity and tongue, is a major contributor to persistent dysgeusia. Epithelial atrophy, basal cell apoptosis, neural damage, and salivary gland dysfunction act synergistically to impair taste perception. Taste changes often begin during the second or third week of treatment and peak between weeks five and seven, with bitter and umami qualities appearing especially vulnerable [54,55,56,57].

Clinically, taste dysfunction in HNC patients is highly heterogeneous, shaped by treatment combinations, tumor location, and individual predispositions. In untreated patients, tumor burden primarily affects sweet and salty taste perception [16,19]. Notably, subjective complaints often show poor correlation with objective testing results, highlighting the complexity of sensory integration and the potential role of psychological factors [29]. Persistent dysgeusia has been associated with higher radiation doses to the oral cavity, concurrent chemotherapy, smoking history, and pre-existing xerostomia [32,45].

Taken together, these alterations extend beyond sensory discomfort, exerting a substantial impact on clinical outcomes. Dysgeusia often leads to reduced food enjoyment, taste aversions, and diminished caloric intake [44], contributing to malnutrition, lower treatment adherence, increased hospitalization rates, and lower quality of life [80]. Quality of life is markedly impaired, as reflected by validated instruments such as the EORTC QLQ-H&N35 and UW-QOL. Psychological consequences—particularly anxiety and depression—are common, compounding the burden of disease and potentially undermining therapeutic compliance [43].

Given these challenges, multiple supportive strategies have been investigated, though high-quality evidence remains limited. Dietary counselling remains the cornerstone of clinical management, offering personalized recommendations on flavor enhancement, texture adaptation, and compensatory techniques based on the specific taste modalities affected [46,51]. Among supplements, zinc sulfate has shown prophylactic potential against radiation-induced dysgeusia [60], while lactoferrin, monosodium glutamate, and miraculin have demonstrated adjunctive efficacy in small studies [63,65]. Non-pharmacological interventions such as photobiomodulation therapy [76], acupuncture [75], and transcutaneous electrical nerve stimulation (TENS) [69] are emerging as promising approaches. Furthermore, surgical techniques aimed at preserving lingual structures may contribute to long-term gustatory preservation [71].

### 4.1. Limitations

Despite growing evidence, the current body of literature exhibits important limitations. Considerable heterogeneity exists across studies in terms of methodology, outcome measures, and patient cohorts. Many investigations rely exclusively on subjective, patient-reported outcomes rather than standardized psychophysical tests. Assessment timing is inconsistent, and sample sizes are frequently underpowered. Moreover, the combined use of chemotherapy and radiotherapy in most HNC protocols complicates causal attribution, rendering interpretation of treatment-specific effects challenging.

Crucially, despite the high prevalence of dysgeusia among HNC patients, this population is often excluded from broader oncologic studies of taste dysfunction. Contributing factors include the complexity of multimodal treatments, the proximity of tumors to gustatory structures, and logistical challenges such as mucositis, dysphagia, and communication barriers that hinder structured sensory testing. However, this exclusion creates a significant knowledge gap in one of the populations most vulnerable to taste-related adverse effects.

While several complementary and integrative strategies show promise in supporting gustatory and salivary function, their efficacy remains to be definitively established. Most current findings should be considered exploratory and interpreted with caution until confirmed by larger, methodologically rigorous randomized controlled trials

### 4.2. Future Directions

Looking ahead, future research should prioritize well-designed longitudinal studies incorporating both objective and patient-reported outcome measures, with validated and reproducible gustatory assessments. Particular attention should be given to the development of standardized taste evaluation tools tailored to head and neck cancer (HNC) populations. There is also a need for HNC-specific clinical trials exploring the efficacy of combination interventions—nutritional, pharmacological, and rehabilitative—stratified by treatment modality and tumor site. Additionally, studies should investigate mechanistic pathways of taste dysfunction, aiming to develop therapeutic strategies that support epithelial integrity, neural preservation, and modulation of the inflammatory response. Ultimately, a multidisciplinary, patient-centered framework will be essential to improve care and long-term outcomes in this vulnerable patient group.

## 5. Conclusions

Taste dysfunction is a prevalent and clinically impactful complication in patients with head and neck cancer, arising from a complex interplay of tumor-related inflammation, cytotoxic treatments, and salivary alterations. This review highlights the multifactorial origins and significant consequences of dysgeusia, which extends beyond sensory discomfort to affect nutritional status, psychological wellbeing, and treatment adherence. Despite its relevance, gustatory dysfunction remains underrepresented in clinical research and is seldom addressed in oncological guidelines. Available management strategies—including dietary counselling, nutritional supplementation, and integrative therapies—show promise but are limited by methodological heterogeneity, small sample sizes, and a lack of standardization. Critically, head and neck cancer patients are often excluded from broader dysgeusia studies due to the complexity of their treatment regimens and anatomical disease sites, creating a notable evidence gap.

## Figures and Tables

**Figure 1 biomedicines-13-01853-f001:**
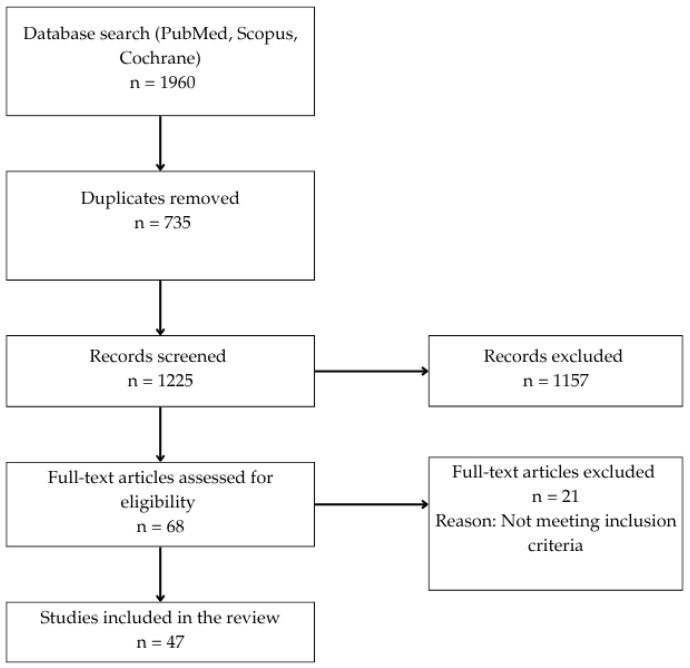
PRISMA flowchart of the study selection process.

**Table 1 biomedicines-13-01853-t001:** Summary of clinical studies on taste dysfunction in head and neck cancer patients related to tumor burden, chemotherapy, or radiotherapy.

Ref.	Study Design	Sample Size (n)	Treatment	Taste Assessment	Results
Singh et al., 2024 [16]	Prospective observational	100 HNC patients	None	NIH Taste Intensity Test (sweet, salty, sour, bitter, umami, baseline only)	Sweet/Salty taste loss correlated with perineural invasion.
Lilja et al., 2018 [17]	Prospective longitudinal	44 HNC patients	Tumor resection + free flap + RT	EGM (electrogustometry) at baseline, 6 weeks, 3 months, 6 months, and 12 months	Persistent taste loss on tumor side; mild recovery at 12 months.
Uí Dhuibhir et al., 2019 [18]	Prospective observational	30 patients with mixed solid tumors, including HNC	None	Taste strips (detection threshold, baseline only) + self-report (taste/smell abnormality)	74% had taste/smell abnormalities; taste more affected; no HNC-specific data.
Cunha et al., 2020 [19]	Cross-sectional observational	31 HNC patients	None	Taste strips (4 concentrations for sweet, salty, and sour and 3 for bitter; 15 points total), baseline only	Hypogeusia in 39%; bitter taste most affected (80.6%).
Abbas et al., 2019 [20]	Cross-sectional observational	59 HNC patients	Surgery ± RT ± CT	UW-QOL v4 (taste domain)	Taste scores significantly lower in advanced-stage tumors (*p* = 0.045).
Tsutsumi et al., 2016 [23]	Prospective observational (molecular focus)	26 HNC patients	CT+ RT	mRNA expression (T1R1, T1R2, T1R3, T2R5) + whole-mouth gustatory test (umami, sweet, bitter)	↓ *T1R3 and ↑ *T2R5 linked to dysgeusia and phantogeusia.
Ihara et al., 2018 [24]	Prospective cohort	30 HNC patients	CRT	gLMS (intensity rating for sweet, salty, sour, bitter; baseline vs. 6 weeks vs. 3 months)	↓ in all tastes at 6 weeks; partial recovery by 3 months.
Epstein et al., 2020 [25]	Prospective longitudinal	10 HNC patients	IMRT ± CT	Edible strips, taste drops, CTCAE v4.0, STTA; baseline vs. 6 weeks post-RT and up to 24 months	Altered taste in all; partial recovery, persistent in some up to 2 years.
Palmieri et al., 2021 [26]	Prospective observational	20 HNC patients	RT + CT	NCI Common Toxicity Criteria (weekly for 6 weeks)	Dysgeusia onset at week 2; peaked at week 5; partial recovery post-treatment.
Galitis et al., 2017 [27]	Longitudinal observational	10 HNC patients	Post-operative RT or chemoradiotherapy	EORTC QLQ-C30 and H&N35 (baseline, end-RT, 3 months post-RT))	Dysgeusia in 88% post-RT; persistent in 50% at 3-month follow-up.
Malta et al., 2021 [28]	Cross-sectional retrospective	514 HNC patients	CT ± RT	CTCAE v5.0 (self-reported dysgeusia grade ≥ 2)	Dysgeusia associated with cisplatin and radiotherapy.
Messing et al., 2021 [29]	Prospective longitudinal	28 HNC patients	RT ± CT ± surgery	Whole-mouth taste test, CiTAS, HNSC (baseline, week 2, week 4, 1, 3, 6 months)	Taste improved by 6 months; persistent dysgeusia correlated with oral dose and xerostomia.
Moroney et al., 2018 [30]	Prospective observational	76 HNC patients	Helical IMRT + CT	CTCAE v4.0 grading for dysgeusia; weekly during RT and post-RT (2, 4, 12 weeks)	Grade 2 dysgeusia in 97.4%; 26.4% had persistent symptoms at 12 weeks.
Sio et al., 2016 [31]	Prospective comparative	81 HNC patients	CT + IMPT or IMRT	MDASI-HN questionnaire (acute, subacute, chronic); baseline	IMPT reduced subacute dysgeusia compared with IMRT.
Chen 2015 [32]	Longitudinal observational	77 HNC patients	RT or CCRT	MSS-moo (taste change assessed at baseline, 4, and 8 weeks during RT)	Taste change peaked at 8 weeks; severity linked to dose, regimen, smoking.
Vempati et al., 2020 [33]	Prospective Phase I trial	34 HNC patients	IMRT + SRS boost + CT	CTCAE v4.03 for toxicity grading; MDASI-HN questionnaire	Acute dysgeusia in 88%; peaked at 9 months; 7% had grade 2 at 24 mo; none grade 3–4.
Asif et al., 2020 [4]	Prospective	21 HNC patients	IMRT/VMAT	Taste strips (4 tastes) and EORTC QoL (baseline, mid-RT, end-RT, 1-, 3-, and 12-month follow-up)	Sweet/salty taste declined during RT; recovered by 1 month.
Barbosa et al., 2019 [34]	Prospective	56 HNC patients	RT	Modified global gustatory test + self-reported qualitative changes (baseline, end-RT, 3 months, 6 months)	Severe taste loss post-RT; recovery by 3–6 months; 14% had qualitative distortions.
Morelli et al., 2023 [35]	Prospective observational	31 HNC patients	IMRT ± CT	CiTAS + EORTC QLQ-C30 + HN43 (baseline, 3 weeks post-RT, 3 months post-RT)	Peak dysgeusia at 3 wks post-RT; correlated with dose to salivary glands.
Riantiningtyas et al., 2024 [15]	Cross-sectional	30 HNC patients	RT ± surgery ± CT	Self-reported sensory perception, oral symptoms, and eating behavior questionnaire (post-treatment)	53% reported altered taste; linked to food aversion and eating difficulties.
Stankevice et al., 2021 [36]	Retrospective observational study	19 HNC patients	RT	TDT, FPT, and EGM for objective gustatory threshold and localization assessment	Persistent dysgeusia reported in two patients treated with RT.
Jin et al., 2020 [37]	Longitudinal	117 HNC patients	IMRT ± surgery ± CT	HNSC checklist for subjective taste change and dietary interference (baseline, mid-RT, post-RT)	Taste change in 93% post-RT; linked to weight loss and dry mouth.
Negi et al., 2017 [38]	Prospective	27 HNC patients	3D-CRT + CT	Forced three-choice stimulus drop technique (weekly during RT, monthly post-RT up to 6 months)	Bitter taste most affected; no full recovery by 6 months.
Martini et al., 2019 [39]	Prospective	31 HNC patients	VMAT RT ± surgery ± CT	CiTAS questionnaire weekly during RT and at 1 week, 1 month, and 6 months post-RT	Taste worsened during RT; phantogeusia improved, but hypogeusia persisted.
Jin et al., 2018 [40]	Longitudinal	114 HNC patients	IMRT ± surgery ± CT	Single-item STA assessment and CiTAS Scale at baseline, mid-treatment, post-treatment, and 1–2 months follow-up	STA in 92% post-RT; only CiTAS “discomfort” sub-scale predicted weight loss.
Kırca & Kutlutürkan, 2016 [41]	Descriptive, Longitudinal	47 HNC patients	RT	MSAS taste item assessed at mid-RT, end-RT, and 1-month post-RT	Taste change is frequent and distressing; associated with dry mouth, sores.
Alfaro et al., 2021 [42]	Cross-sectional	40 HNC patients	RT ± CT ± surgery	Regional (tip of the tongue) and whole-mouth taste tests using the gLMS Scale; cross-sectional assessment	Localized taste dysfunction detected at the tip of the tongue (sweet, salty, bitter stimuli) despite preserved whole-mouth taste perception.
Mathlin et al., 2023 [43]	Prospective	61 HNC patients	VMAT RT ± CT	MDASI-HN questionnaire at week 1 and week 4 of RT; supplementary coping questions for dysgeusia at week 4	97% reported taste changes; worse with chewing; more frequent in females.
Alvarez-Camacho et al., 2016 [44]	Longitudinal	160 HNC patients	RT ± CT ± surgery	Self-report: CCS + UW-QoL (pre, post, 2.5 mo)	Taste/Smell changes predicted worse QoL post-treatment.
Sapir 2016 [45]	Prospective longitudinal	73 HNC patients	CRT via IMRT	Patient-reported taste (HNQOL, UWQOL) + unstimulated/stimulated salivary flow; baseline to 12 months	Dysgeusia in 50% (1 mo), 23% (12 months); correlated with oral cavity dose.

Legends: NIH = National Institutes of Health; EGM = electrogustometry; UW-QOL = University of Washington Quality of Life Questionnaire; CRT = chemoradiotherapy; gLMS = Generalized Labeled Magnitude Scale; CTCAE = Common Terminology Criteria for Adverse Events; STTA = Subjective Taste and Temperature Alteration; EORTC QLQ-C30 = European Organisation for Research and Treatment of Cancer Quality of Life Questionnaire—Core 30; H&N35 / HN43 = EORTC Head and Neck Modules; CiTAS = Chemotherapy-induced Taste Alteration Scale; HNSC = Head and Neck Symptom Checklist; MSS-moo = Memorial Symptom Scale—Modified for Oral Oncology; TDT = Taste Drop Test; FPT = Filter Paper Test; STA = subjective taste alteration; MSAS = Memorial Symptom Assessment Scale; CCS = Chemotherapy Convenience Scale; HNQOL = Head and Neck Quality of Life Questionnaire; D50 = dose to 50% of the oral cavity volume. * Note: ↑ indicates an increase; ↓ indicates a decrease in the corresponding variable or clinical outcome.

**Table 2 biomedicines-13-01853-t002:** Summary of clinical studies investigating nutritional and supportive interventions for taste dysfunction in head and neck cancer patients.

Ref.	Study Design	Sample Size (n)	Intervention/Treatment	Outcome Measures	Results
Shono et al. (2021) [63]	Randomized controlled cohort study (non-blinded)	51 HNC patients	Monosodium glutamate (MSG) (2.7 g/day) during chemoradiotherapy	T1R3 gene expression, VAS for dysgeusia, daily energy intake	MSG preserved T1R3 expression, improved taste and calorie intake.
López-Plaza et al. (2023) [65]	Randomized, placebo-controlled, triple-blind trial (pilot)	31 malnourished cancer patients (HNC included)	Miraculin-based supplement (standard/high dose) vs. placebo	Taste acuity (electrogustometry), dietary intake, QoL	Standard dose improved taste, intake, QoL; no adverse events.
Khan et al. (2019) [60]	Double-blind randomized controlled trial	70 HNC patients	Zinc sulphate 50 mg TID vs. placebo during and after CCRT	Detection and recognition thresholds for 4 tastes	No significant benefit overall; some improvement in sweet and sour recognition.
Ben-Arye et al. (2018) [66]	Prospective chart-based study	34 (some patients with HNC, not quantified)	Complementary medicine: sage, carob, wheatgrass, acupuncture, mind–body therapies	ESAS, MYCAW for symptom improvement	85% reported taste improvement; herbal/acupuncture was the most beneficial.
Lesser et al. (2022) [62]	Pilot clinical trial	26 not quantified, but HNC patients included	Lactoferrin 750 mg/day for 30 days	TSQ (taste, smell, composite scores)	Significant taste/smell improvement at 60 days; partial at 30.
Heiser et al. (2016) [67]	Prospective cohort study (pre-post)	98 HNC patients	Liposomal spray (oral and nasal) for 2 months	Taste strips, smell test, xerostomia questionnaire	Significant improvement in taste, smell, and xerostomia symptoms.
Epstein et al. (2019) [68]	Clinical case series	14 (9 HNC)	Zinc, clonazepam, megestrol acetate, dronabinol, PBMT	STTA, CTCAE, chemical gustometry	71% of patients reported improvement in taste function
Dalbem Paim et al. (2019) [69]	Randomized controlled trial	68 HNC patients	Transcutaneous Electrical Nerve Stimulation (TENS), 8 sessions	Stimulated salivary flow (SSF), VAS for salivation, QoL (UW-QOL)	TENS significantly improved SSF, self-perceived saliva flow, and QoL up to 6 months.
Feng et al., 2019 [70]	Prospective	60 HNC patients	IMRT ± CT; bite block	Clinical observation (presence of dysgeusia at end-RT; with vs. without bite block)	Bite block prevented dysgeusia; reduced mucosal dose.
Lu et al., 2020 [71]	Prospective case series	21 HNC patients	Surgical resection + modified anterior–posterior tongue rotation flap	UW-QOL v4 (self-reported taste domain; 12–24 months follow-up)	All patients reported normal taste post-op. Flap preserved tongue length and symmetry. Excellent outcomes also for swallowing, chewing, and speech.
Li et al., 2016 [72]	Retrospective comparative study	41 HNC patients	Surgical resection + reconstruction with RFFF or PMMF	UW-QOL v4 (self-reported taste domain; ≥12 months follow-up)	No significant difference in taste function between RFFF and PMMF (*p* = 0.673).
Yuan et al., 2016 [73]	Prospective observational study	67 HNC patients	Surgical resection + reconstruction with ALTFF or RFFF	UW-QOL v4 (self-reported taste domain; 6- and 12-months follow-up)	Taste improved at 12 months; no difference between flaps.
Yue et al., 2018 [74]	Prospective observational study	139 HNC patients	Tumor resection ± immediate reconstruction with free flap	UW-QOL v4 (self-reported taste domain; ≥12 months follow-up)	Taste was among the worst domains; no difference by treatment group.
Djali et al., 2020 [75]	Case report	65-year-old man with stage I laryngeal SCC post-RT	Acupuncture (body points, auricular battlefield acupuncture, wrist balancing method); 12 sessions, 2×/week	Clinical observation (VAS)	Full taste recovery and pain reduction (VAS 4 → 1) after 12 sessions.
El Mobadder et al., 2019 [76]	Case series	3 cancer patients with different diagnoses; 1 HNC patient.	Photobiomodulation therapy (635 nm diode laser; 10 sessions on tongue dorsum and lateral surfaces)	ISO 3972:2011 (sip and spit test for 5 basic tastes)	Taste score improved from 0/5 to 5/5 after 10 PBM sessions.
Yangchen et al., 2016 [77]	Pilot controlled cohort study (not RCT)	24 HNC patients	Cerrobend shielding stent during radiotherapy vs. no stent (control)	RTOG 0435 Scale: multiple oral side effects including taste alteration assessed at 1 and 3 months	No significant difference in taste alteration between groups.
Fernandes et al., 2022 [78]	Double-blind RCT	60 HNC Patients	Brazilian organic propolis spray vs. placebo (6×/day, during RT)	NCI CTCAE (weekly dysgeusia score)	Lower dysgeusia in propolis group (not significant); ↓ candidiasis, IL-1β, TNF-α.

Legend: MSG = Monosodium glutamate; VAS = Visual Analog Scale; TID = ter in die (three times per day); CCRT = concurrent chemoradiotherapy; ESAS = Edmonton Symptom Assessment Scale; MYCAW = Measure Yourself Concerns and Wellbeing; TSQ = Taste and Smell Questionnaire; SSF = stimulated salivary flow; PBMT = photobiomodulation therapy; RFFF = radial forearm free flap; PMMF = pectoralis major myocutaneous flap; ALTFF = anterolateral thigh free flap; ISO 3972:2011 = International Standard for Sensory Analysis—Methodology for Basic Taste Recognition; RTOG 0435 = Radiation Therapy Oncology Group protocol scale for head and neck toxicity; NCI CTCAE = National Cancer Institute—Common Terminology Criteria for Adverse Events. ↓ indicates a decrease in the corresponding variable or clinical outcome.

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
