# Peer review of "Taste Dysfunction in Head and Neck Cancer: Pathophysiology and Clinical Management—A Comprehensive Review"

_biomedicines, 2025, doi:10.3390/biomedicines13081853_

Round 1
Reviewer 1 Report
Comments and Suggestions for Authors
Dear authors, please find attached

Author Response
Dear Reviewer,
We sincerely thank you for your thoughtful and constructive feedback. Your comments have been extremely helpful in improving the quality and clarity of our manuscript. Please find below our point-by-point responses:
Introduction: we have revised the Introduction section to ensure a more cohesive and narrative progression. The updated version integrates the relevant background and rationale more fluidly, avoiding a segmented or bullet-point format.
Methods: The Methods section has been revised to explicitly indicate adherence to the PRISMA-ScR (Preferred Reporting Items for Systematic Reviews and Meta-Analyses Extension for Scoping Reviews) guidelines [Tricco et al., 2018], which better reflect the scoping nature of our work. Additionally, we now reference the PCC (Population–Concept–Context) model that guided both our eligibility criteria and data extraction. To ensure full transparency, we have added Appendix A with the complete PubMed search strategy, including MeSH terms, free-text keywords, Boolean operators, and filters used.
Results: we have substantially revised the Results section by reducing redundancy and condensing descriptive passages. The revised section now closely aligns with the corresponding table, using it to guide the interpretation of key findings and avoiding repetition. We believe these changes improve readability and better highlight clinically relevant data.
Discussion: As suggested, we have included a dedicated subsection titled “4.1 Limitations” within the Discussion. This new section outlines the primary limitations of the included literature and of our review, including heterogeneity of study design and outcome measures, small sample sizes, reliance on subjective patient-reported outcomes, and the frequent exclusion of HNC patients from broader oncologic studies
References: The requested references have been added in lines 561–576 (note that the number lines is changed due to text revision) to support the mechanistic explanations regarding chemotherapy- and radiotherapy-induced taste dysfunction.
We are grateful for your time and effort in reviewing our manuscript.
Reviewer 2 Report
Comments and Suggestions for Authors
Dear Authors,
This paper is well written and provides a comprehensive literature review.
However, it doesn't offer any new insights. Nevertheless, it gives to the reader a global overview of taste and smell problems in a patients with a H&N carcinoma, particularly in patients oral cavity carcinoma.
Author Response
Dear Reviewer,
Thank you for your valuable comments and for recognizing the overall clarity and comprehensive nature of our manuscript.
As a comprehensive review, our primary aim was to synthesize and critically appraise recent evidence (2015–2025), with a specific focus on taste dysfunction in head and neck cancer patients. While we did not present original experimental data, we believe our work contributes meaningfully by:
-Providing an updated, structured synthesis of clinical evidence, categorized by etiology (tumor-related, chemotherapy-induced, radiotherapy-induced);
-Emphasizing gaps in clinical management, including the absence of standardized interventions specific to taste dysfunction in HNC patients;
-Including studies specifically addressing taste dysfunction in HNC, thereby increasing the specificity of the review.
We appreciate your feedback and hope these clarifications address your concerns.
Reviewer 3 Report
Comments and Suggestions for Authors
The review addresses an important and under-recognized complication in patients with head and neck cancer (HNC)—taste dysfunction. The manuscript provides a broad overview of the current evidence; however, it would benefit from enhanced methodological rigor, improved structural coherence, and careful language refinement.
My detailed comments are as follows:
-
Surgical Resection as an Independent Cause of Dysgeusia: While the manuscript briefly touches on taste outcomes in the context of reconstructive procedures, it does not adequately address surgical resection itself as a primary and significant contributor to taste dysfunction. A more focused discussion on the anatomical and neural consequences of surgery—particularly involving structures such as the anterior tongue, soft palate, and floor of the mouth—would considerably improve the clinical depth of the review.
-
Quantitative Synthesis and Prevalence Reporting: The authors note that meta-analysis was not feasible due to heterogeneity; however, the review currently lacks even basic quantitative synthesis. I recommend including descriptive statistics such as the proportion of patients reporting dysgeusia by treatment modality (e.g., surgery, chemotherapy, radiotherapy), and summarizing trends across interventions. Additionally, the abstract and results section would be strengthened by reporting the overall prevalence range of taste dysfunction (e.g., “reported prevalence ranged from 17% to 96%”).
-
Language and Terminology Issues: There are several typographical and grammatical errors that need attention. For example, “disfuntion” should be corrected to “dysfunction,” and phrases such as “partially or totally lost of taste food” are awkward and incorrect. A clearer, more accurate phrasing would be: “classified as dysgeusia (altered taste) or ageusia (complete loss of taste).” A thorough language edit is recommended to improve clarity.
-
Pharmacological Treatments Section: This section is notably underdeveloped compared to the rest of the manuscript. I suggest elaborating on the limitations of existing pharmacologic approaches, such as the lack of randomized controlled trials (RCTs), heterogeneity in outcome measures, and issues with patient adherence.
-
Future Research Directions: While the discussion is strong, it would benefit from a clearly demarcated subsection outlining "Future Research Directions." This could include suggestions such as the development of standardized gustatory assessment tools, the need for HNC-specific clinical trials, exploration of combination interventions, and the incorporation of objective and patient-reported outcome measures in future studies.
Comments on the Quality of English Language
needs editing
Author Response
Dear Reviewer,
We are grateful for your thoughtful and constructive feedback. Below, we address each of your comments point by point.
Surgical Resection as an Independent Cause of Dysgeusia: We have added a new subsection titled “3.1.1 Surgical Resection as an Independent Contributor to Taste Dysfunction” within the broader section “3.1 Taste Dysfunction Caused by Cancer Itself.” This revised subsection discusses the anatomical and neural consequences of resections involving gustatory structures such as the anterior two-thirds of the tongue, the floor of the mouth, and the soft palate. We believe this addition substantially enhances the clinical scope of the review, in accordance with your recommendation.
Quantitative Synthesis and Prevalence Reporting: We appreciate your suggestion to improve the quantitative dimension of the manuscript. In response, we have included a descriptive synthesis in the final paragraph of Section 3.3, summarizing the prevalence of dysgeusia across different treatment modalities. As further recommended, we have explicitly reported the overall prevalence range of taste dysfunction (from 39% to 97.4%) in both the Abstract and Discussion sections.
Language and Terminology Issues: We have carefully revised the manuscript to correct typographical and grammatical issues, including the correction of “disfuntion” to “dysfunction.” Moreover, the entire manuscript has undergone a comprehensive English language revision to improve clarity, accuracy, and fluency. We believe these refinements contribute significantly to the readability and professional tone of the manuscript.
Pharmacological Treatments Section: We are grateful for your suggestion to strengthen this section. In response, we have expanded the Pharmacological Treatments subsection to provide a more critical appraisal of current evidence.
Future Research Directions: As suggested, we have added a dedicated subsection titled “4.2 Future Directions” at the end of the Discussion. This subsection outlines priority areas for future investigation.
We thank you sincerely for your invaluable input, which has greatly contributed to strengthening the scientific rigor and overall quality of our manuscript.
Round 2
Reviewer 3 Report
Comments and Suggestions for Authors
the manuscript was improved.